# UDAPDR: Unsupervised Domain Adaptation via LLM Prompting and Distillation of Rerankers

**Jon Saad-Falcon**[1]  **Omar Khattab**[1]  **Keshav Santhanam**[1]  **Radu Florian**[2]  **Martin Franz**[2]
**Salim Roukos**[2]  **Avirup Sil**[2]  **Md Arafat Sultan**[2]  **Christopher Potts**[1]

[1]**Stanford University**   [2]**IBM Research AI**

## Abstract

Many information retrieval tasks require large labeled datasets for fine-tuning. However, such datasets are often unavailable, and their utility for real-world applications can diminish quickly due to domain shifts. To address this challenge, we develop and motivate a method for using large language models (LLMs) to generate large numbers of synthetic queries cheaply. The method begins by generating a small number of synthetic queries using an expensive LLM. After that, a much less expensive one is used to create large numbers of synthetic queries, which are used to fine-tune a family of reranker models. These rerankers are then distilled into a single efficient retriever for use in the target domain. We show that this technique boosts zero-shot accuracy in long-tail domains and achieves substantially lower latency than standard reranking methods.

## 1 Introduction

The advent of neural information retrieval (IR) has led to notable performance improvements on document and passage retrieval tasks (Nogueira and Cho, 2019; Khattab and Zaharia, 2020; Formal et al., 2021) as well as downstream knowledge-intensive NLP tasks such as open-domain question-answering and fact verification (Guu et al., 2020; Lewis et al., 2020; Khattab et al., 2021; Izacard et al., 2022). Neural retrievers for these tasks often benefit from fine-tuning on large labeled datasets such as SQuAD (Rajpurkar et al., 2018), Natural Questions (NQ) (Kwiatkowski et al., 2019), and KILT (Petroni et al., 2021). However, IR models can experience significant drops in accuracy due to distribution shifts from the training to the target domain (Thakur et al., 2021; Santhanam et al., 2022b). For example, dense retrieval models trained on MS MARCO (Nguyen et al., 2016) might not generalize well to queries about COVID-19 scientific publications (Voorhees et al., 2021; Wang et al.,

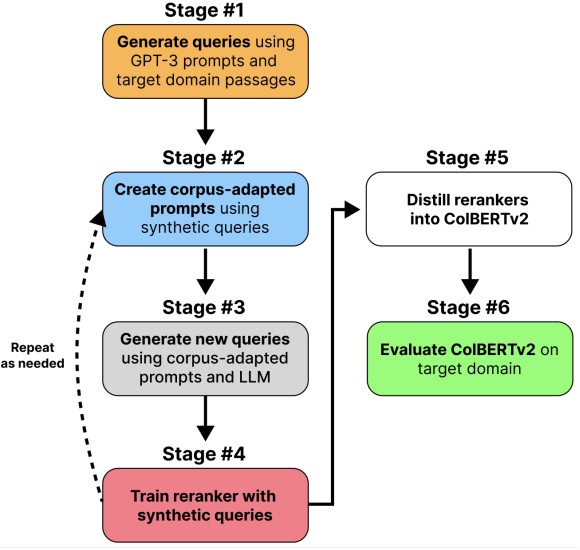

Figure 1: Overview of UDAPDR. An expensive LLM like GPT-3 is used to create an initial set of synthetic queries. These are incorporated into a set of prompts for a less expensive LLM that can generate large numbers of synthetic queries cheaply. The queries stemming from each prompt are used to train separate rerankers, and these are distilled into a single ColBERTv2 retriever for use in the target domain.

2020), considering for instance that MS MARCO predates COVID-19 and thus lacks related topics.

Recent work has sought to adapt IR models to new domains by using large language models (LLMs) to create synthetic target-domain datasets for fine-tuning retrievers (Bonifacio et al., 2022; Meng et al., 2022; Dua et al., 2022). For example, using synthetic queries, Thakur et al. (2021) and Dai et al. (2022) fine-tune the retriever itself and train a cross-encoder to serve as a passage reranker for improving retrieval accuracy. This significantly improves retriever performance in novel domains, but it comes at a high computational cost stemming from extensive use of LLMs. This has limited the applicability of these methods for researchers and practitioners, particularly in high-demand, user-

facing settings.

In this paper, we develop **U**nsupervised **D**omain **A**daptation via LLM **P**rompting and **D**istillation of **R**erankers (UDAPDR),[1] an efficient strategy for using LLMs to facilitate unsupervised domain adaptation of neural retriever models. We show that UDAPDR leads to large gains in zero-shot settings on a diverse range of domains.

The approach is outlined in Figure 1. We begin with a collection of passages from a target domain (no in-domain queries or labels are required) as well as a prompting strategy incorporating these passages with the goal of query generation. A powerful (and perhaps expensive) language model like GPT-3 is used to create a modest number of synthetic queries. These queries form the basis for corpus-adapted prompts that provide examples of passages with good and bad queries, with the goal of generating good queries for new target domain passages. These prompts are fed to a smaller (and presumably less expensive) LM that can generate a very large number of queries for fine-tuning neural rerankers. We train a separate reranker on the queries from each of these corpus-adapted prompts, and these rerankers are distilled into a single student ColBERTv2 retriever (Khattab and Zaharia, 2020; Santhanam et al., 2022b,a), which is evaluated on the target domain.

By distilling from multiple passage rerankers instead of a single one, we improve the utility of ColBERTv2, preserving more retrieval accuracy gains while reducing latency at inference. Our core contributions are as follows:

- We propose UDAPDR, a novel unsupervised domain adaptation method for neural IR that strategically leverages expensive LLMs like GPT-3 (Brown et al., 2020) and less expensive ones like Flan-T5 XXL (Chung et al., 2022), as well as multiple passage rerankers. Our approach improves retrieval accuracy in zero-shot settings for LoTTE (Santhanam et al., 2022b), SQuAD, and NQ.

- We preserve the accuracy gains of these rerankers while maintaining the competitive latency of ColBERTv2. This leads to substantial reductions in query latency.

- Unlike a number of previous domain adaptation approaches that utilize millions of synthetic queries, our technique only requires

1000s of synthetic queries to prove effective and is compatible with various LLMs designed for handling instruction-based tasks like creating synthetic queries (e.g., GPT-3, T5, Flan-T5).

- We generate synthetic queries using multiple prompting strategies that leverage GPT-3 and Flan-T5 XXL. This bolsters the effectiveness of our unsupervised domain adaptation approach. The broader set of synthetic queries allows us to fine-tune multiple passage rerankers and distill them more effectively.

## 2 Related Work

### 2.1 Data Augmentation for Neural IR

LLMs have been used to generate synthetic datasets (He et al., 2022; Yang et al., 2020; Anaby-Tavor et al., 2020; Kumar et al., 2020), which have been shown to support effective domain adaptation in Transformer-based architectures (Vaswani et al., 2017) across various tasks. LLMs have also been used to improve IR accuracy in new domains via the creation of synthetic datasets for retriever fine-tuning (Bonifacio et al., 2022; Meng et al., 2022).

Domain shift is the most pressing challenge for domain transfer. Dua et al. (2022) categorize different types of domain shifts, such as changes in query or document distributions, and provide intervention strategies for addressing each type of shift using synthetic data and indexing strategies.

Query generation can help retrieval models trained on general domain tasks adapt to more targeted domains through the use of generated query–passage pairs (Ma et al., 2020; Nogueira et al., 2019). Wang et al. (2022) also use generative models to pseudo-label synthetic queries, using the generated data to adapt dense retrieval models to domain-specific datasets like BEIR (Thakur et al., 2021). Thakur et al. (2021) and Dai et al. (2022) generate millions of synthetic examples for fine-tuning dense retrieval models, allowing for zero-shot and few-shot domain adaptation.

Synthetic queries can also be used to train passage rerankers that assist neural retrievers. Cross-encoders trained with synthetic queries boost retrieval accuracy substantially while proving more robust to domain shifts (Thakur et al., 2020, 2021; Humeau et al., 2019). Dai et al. (2022) explore training the *few-shot* reranker Promptagator++, leveraging an unsupervised domain-adaptation approach that utilizes millions of synthetically gener-

---

[1]pronounced: Yoo-Dap-ter

ated queries to train a passage reranker alongside a dense retrieval model. Additionally, Wang et al. (2022) found using zero-shot cross-encoders for reranking could further improve quality.

However, due to the high computational cost of rerankers at inference, both Dai et al. (2022) and Wang et al. (2022) found it unlikely these approaches would be deployed in user-facing settings for information retrieval. Our work seeks to bolster the utility of passage rerankers in information retrieval systems. Overall, Dai et al. (2022) and Wang et al. (2022) demonstrated the efficacy of unsupervised domain adaptation approaches utilizing synthesized queries for fine-tuning dense retrievers or passage rerankers. By using distillation strategies, we can avoid their high computational cost while preserving the latent knowledge gained through unsupervised domain adaptation approaches.

## 2.2 Pretraining Objectives for IR

Pretraining objectives can help neural IR systems adapt to new domains without annotations. Masked Language Modeling (MLM) (Devlin et al., 2019) and Inverse Cloze Task (ICT) (Lee et al., 2019) offer unsupervised approaches for helping retrieval models adapt to new domains. Beyond MLM and ICT, Chang et al. (2020) proposed two unsupervised pretraining tasks, Body First Selection (BFS) and Wiki Link Prediction (WLP), which use sampled in-domain sentences and passages to warm-up a neural retriever to new domains. Additionally, Gysel et al. (2018) developed the Neural Vector Space Model (NVSM), an unsupervised pretraining task for news article retrieval that utilizes learned low-dimensional representations of documents and words. Izacard et al. (2021) also explore a contrastive learning objective for unsupervised training of dense retrievers, improving retrieval accuracy in new domains across different languages.

These pretraining objectives can be paired with additional domain adaptation strategies. Wang et al. (2022) coupled ICT with synthetic query data to achieve domain adaptation in dense retrieval models without the need for annotations. Dai et al. (2022) also paired the contrastive learning objective in Izacard et al. (2021) with their unsupervised Promptagator strategy. While our zero-shot domain adaptation approach can pair with other techniques, it does not require any further pretrainingfor bolstered retrieval performance; our approach only needs the language model pretraining of the re-

triever's base model (Devlin et al., 2019), and we show that it combines effectively with multi-vector retrievers (Khattab and Zaharia, 2020; Santhanam et al., 2022b).

## 3 Methodology

Figure 1 outlines each stage of the UDAPDR strategy. For the target domain $T$, our approach requires access to in-domain passages (i.e., within the domain of $T$). However, it does not require any in-domain queries or labels. The overall goal is to leverage our store of in-domain passages and LLM prompting to inexpensively generate large numbers of synthetic queries for passages. These synthetic queries are used to train domain-specific reranking models that serve as teachers for a single retriever. The specific stages of this process are as follows:

**Stage 1:** We begin with a set of prompts that embed examples of passages paired with good and bad queries and that seek to have the model generate a novel good query for a new passage. We sample $X$ in-domain passages from the target domain $T$, and we use the prompts to generate $5X$ synthetic queries. In our experiments, we test values of $X$ such as 5, 10, 50, and 100. (In Appendix A, we explore different strategies for selecting the $X$ in-domain passages from the target domain.)

In this stage, we use GPT-3 (Brown et al., 2020), specifically the text-davinci-002 model. The guiding idea is to use a very effective LLM for the first stage, to seed the process with very high quality queries. We employ the five prompting strategies in Figure 2. Two of our prompts are from Bonifacio et al. 2022, where they proved successful for generating synthetic queries in a few-shot setting. The remaining three use a zero-shot strategy and were inspired by prompts in Asai et al. 2022.

**Stage 2:** The queries generated in Stage 1 form the basis for prompts in which passages from the target domain $T$ are paired with good and bad synthetic queries. The prompt seeks to lead the model to generate a good query for a new passage. Our prompt template for this stage is given in Figure 3. We create $Y$ corpus-adapted prompts in this fashion, which vary according to the demonstrations they include. In our experiments, we test out several values for $Y$, specifically, 1, 5, and 10. This *programmatic* creation of few-shot demonstrations for language models is inspired by the Demonstrate stage of the DSP framework (Khattab et al., 2022).

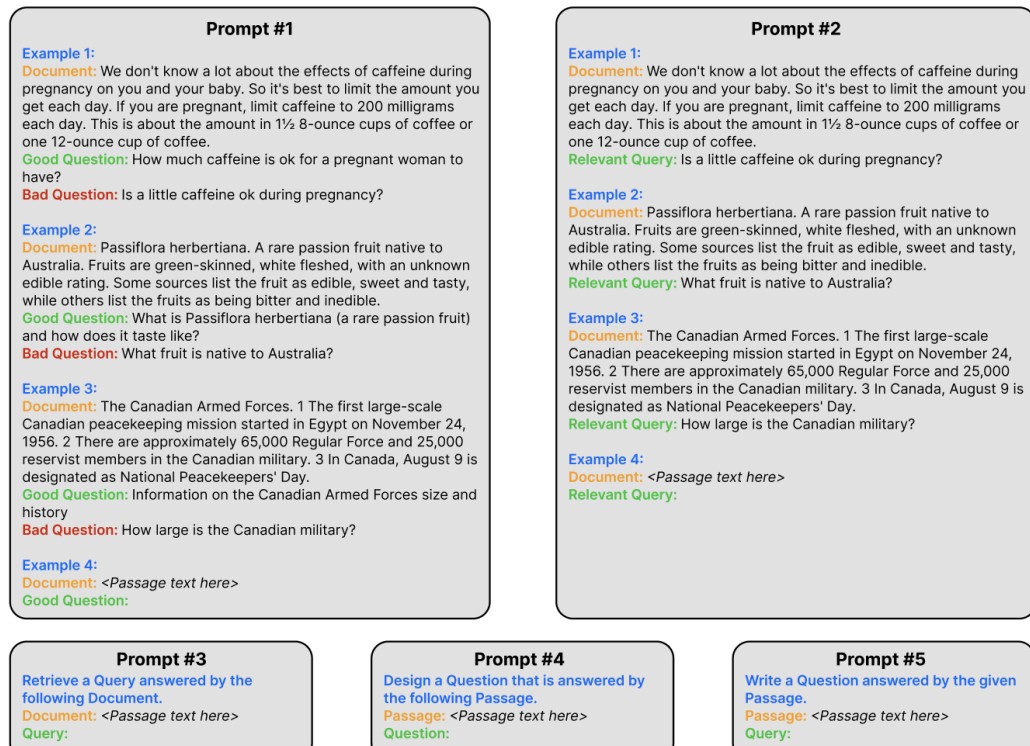

Figure 2: The five prompts used in Stage 1 (Section 3). The few-shot prompts #1 and #2 were inspired by Bonifacio et al. (2022) while the zero-shot prompts #3, #4, and #5 were inspired by Asai et al. (2022). In our experiments, we prompt GPT-3 in this stage to generate an initial set of queries.

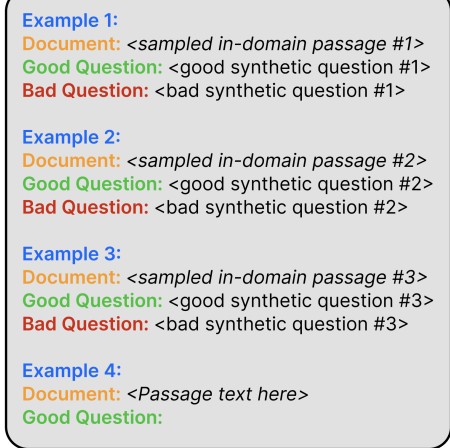

Figure 3: The prompt template used in Stage 2. (Section 3). In our experiments, we create $Y$ variants of this prompt, and each one is used with Flan-T5 XXL to generate $Z$ queries for each $Y$.

**Stage 3:** Each of the corpus-adapted prompts from Stage 2 is used to generate a unique set of $Z$ queries with Flan-T5 XXL (Chung et al., 2022). The gold passage for each synthetic query is the passage it was generated from. We have the option of letting $Z$ be large because using Flan-T5 XXL is considerably less expensive than using GPT-3 as

we did in Stage 1. In our experiments, we test 1K, 10K, 100K, and 1M as values for $Z$. We primarily focus on $Z = 10K$ and 100K in Section 4.3.

We use multiple corpus-adapted prompts to mitigate edge cases in which we create a low-quality prompt based on the chosen synthetic queries and in-domain passages from the target domain $T$. (See Stage 4 below for a description of how low-quality prompts can optionally be detected and removed.)

As a quality filter for selecting synthetic queries, we test whether a synthetic query can return its gold passage within the top 20 retrieved results using a zero-shot ColBERTv2 retriever. We only use synthetic queries that pass this filter, which has been shown to improve domain adaptation in neural IR (Dai et al., 2022; Jeronymo et al., 2023).

**Stage 4:** With each set of synthetic queries generated using the $Y$ corpus-adapted prompts in Stage 3, we train an individual passage reranker from scratch for the target domain $T$. These will be used as teachers for a single ColBERTv2 model in Stage 5. Our multi-reranker strategy draws inspiration from Hofstätter et al. (2021), who found a teacher ensemble effective for knowledge distillation into retrievers. At this stage, we can simply

use all $Y$ of these rerankers for the distillation process. As an alternative, we can select the $N$ best rerankers based on their accuracy on the validation set of the target domain. For our main experiments, we use all of these rerankers. This is the most general case, in which we do *not* assume that a validation set exists for the target domain. (In Appendix A, we evaluate settings where a subset of them is used.)

**Stage 5:** The domain-specific passage rerankers from Stage 4 serve as multi-teachers creating a single ColBERTv2 retriever in a multi-teacher distillation process. For distillation, we use annotated triples that are created by using the trained domain-specific reranker to label additional synthetic questions. Overall, our distillation process allows us to improve the retrieval accuracy of ColBERTv2 without needing to use the rerankers at inference.

**Stage 6:** We test our domain-adapted ColBERTv2 retriever on the evaluation set for the target domain $T$. This corresponds to deployment of the retriever within the target domain.

# 4 Experiments

## 4.1 Models

We leverage the Demonstrate-Search-Predict (DSP) (Khattab et al., 2022) codebase for running our experiments. The DSP architecture allows us to build a modular system with both retrieval models and LLMs. For our passage rerankers, we selected DeBERTaV3-Large (He et al., 2021) as the cross-encoder after performing comparison experiments with RoBERTa-Large (Liu et al., 2019), BERT (Devlin et al., 2019), and MiniLMv2 (Wang et al., 2021). For our IR system, we use the ColBERTv2 retriever since it remains competitive for both accuracy and query latency across various domains and platforms (Santhanam et al., 2022c).

## 4.2 Datasets

For our experiments, we use LoTTE (Santhanam et al., 2022b), BEIR (Thakur et al., 2021), NQ (Kwiatkowski et al., 2019), and SQuAD (Rajpurkar et al., 2016). This allows us to cover both long-tail IR (LoTTE, BEIR) and general-knowledge question answering (NQ, SQuAD).

We note that NQ and SQuAD were both part of Flan-T5's pretraining datasets (Chung et al., 2022). Wikipedia passages used in NQ and SQuAD were also part of DeBERTaV3 and GPT-3's pretraining

datasets (He et al., 2021; Brown et al., 2020). Similarly, the raw StackExchange answers and questions (i.e., from which LoTTE-Forum is derived) may overlap in part with the training data of GPT-3. The overlap between pretraining and evaluation datasets may impact the efficacy of domain adaptation on NQ and SQuAD, leading to increased accuracy not caused by our approach.

## 4.3 Multi-Reranker Domain Adaptation

Table 1 provides our main results for UDAPDR accuracy. For these experiments, we set a total budget of 100K synthetic queries and distribute these equally across the chosen number of rerankers to be used as teachers in the distillation process. When exploring UDAPDR system designs, we report dev results, and we report core test results for LoTTE and BEIR in Section 4.7.

We compare UDAPDR to two baselines. In the leftmost column of Table 1, we have a Zero-shot ColBERTv2 retriever (no distillation). This model has been shown to be extremely effective in our benchmark domains, and it is very low latency, so it serves as an ambitious baseline. In the rightmost column, we have a Zero-shot ColBERTv2 retriever paired with a single non-distilled passage reranker, trained on 100K synthetic queries. We expect this model to be extremely competitive in terms of accuracy, but also very high latency.

All versions of UDAPDR are far superior to Zero-shot ColBERTv2 across all the domains we evaluated. In addition, two settings of our model are competitive with or superior to Zero-shot ColBERTv2 plus a Reranker: distilling into ColBERTv2 the scores from 5 rerankers, each trained on 20k synthetic queries, as well as 10 rerankers, each trained on 10k synthetic queries.

## 4.4 Query Latency

The accuracy results in Table 1 show that UDAPDR is highly effective. In addition to this, Table 2 reports a set of *latency* evaluations using the LoTTe Lifestyle section. Our latency costs refer to the complete retrieval process for a single query, from query encoding to the last stage of passage retrieval.

Zero-shot ColBERTv2 is known to have low retrieval latency (Santhanam et al., 2022a). However, its accuracy (repeated from Table 1), which is at a state-of-the-art level (Santhanam et al., 2022c), trails by large margins the methods we propose in this work. UDAPDR (line 2) has similar latency but also achieves the best accuracy results. The

| | ColBERTv2 Distillation with UDAPDR | | | |
| | Zero-shot ColBERTv2 | $Y = 1$ reranker, $Z = 100$k queries | $Y = 5$ rerankers, $Z = 20$k queries | $Y = 10$ rerankers, $Z = 10$k queries | Zero-shot ColBERTv2 + Reranker |
|---|---|---|---|---|---|
| LoTTE Lifestyle | 64.5 | 73.0 | **74.8** | 74.4 | 73.5 |
| LoTTE Techology | 44.5 | 50.2 | **51.3** | 51.1 | 50.6 |
| LoTTE Writing | 80.0 | 84.3 | 85.7 | **86.2** | 85.5 |
| LoTTE Recreation | 70.8 | 76.9 | **80.4** | 79.8 | 79.1 |
| LoTTE Science | 61.5 | 65.6 | 67.9 | **68.0** | 67.2 |
| LoTTE Pooled | 63.7 | 70.0 | 72.1 | **72.2** | 71.1 |
| NaturalQuestions | 68.9 | 72.4 | 73.7 | **74.0** | 73.9 |
| SQuAD | 65.0 | 71.8 | **73.8** | 73.6 | 72.6 |

Table 1: **Success@5 for Multi-Reranker Domain Adaptation Strategies with Different Synthetic Query Counts.** LoTTE results are for the Forum configuration. All results are for dev sets. The reranker used is DeBERTa-v3-Large. The ColBERTv2 distillation strategies train $Y$ rerankers each with $Z$ synthetic queries before distilling each reranker with the same ColBERTv2 model. No selection process for rerankers is needed nor access to annotated in-domain dev sets (cf. Table 7 in our Appendices). The non-distilled reranker in the final column is trained on 100K synthetic queries created using Flan-T5 XXL model and the prompting strategy outlined in Section 3.

| Retriever and Reranker | Passages Reranked | Query Latency | Success@5 |
|---|---|---|---|
| Zero-shot ColBERTv2 | N/A | 35 ms | 64.5 |
| ColBERTv2 Distillation: $Y = 5$ rerankers, $Z = 20$k queries | N/A | 35 ms | 74.8 |
| Zero-shot ColBERTv2 + Reranker | 20 | 412 ms | 73.3 |
| Zero-shot ColBERTv2 + Reranker | 100 | 2060 ms | 73.5 |
| Zero-shot ColBERTv2 + Reranker | 1000 | 20600 ms | 73.5 |

Table 2: **Average Single Query Latency for Retrieval + Reranker Systems.** Latencies and Success@5 are for LoTTE Lifestyle. The ColBERTv2 distillation strategies train $Y$ rerankers each with $Z$ synthetic queries before distilling each reranker with the same ColBERTv2 model. These experiments were performed on a single NVIDIA V100 GPU with PyTorch, version 1.13 (Paszke et al., 2019). Query latencies rounded to three significant digits.

Zero-shot ColBERTv2 + Reranker models come close, but only with significantly higher latency.

## 4.5 Impact of Pretrained Components

UDAPDR involves three pretrained components: GPT-3 to generate our initial set of synthetic queries, Flan-T5 XXL to generate our second, larger set of synthetic queries, and DeBERTaV3-Large for the passage rerankers. What is the impact of these specific components on system behavior?

To begin to address this question, we explored a range of variants. These results are summarized in Table 4. Our primary setting for UDAPDR performs the best, but it is noteworthy that very competitive performance can be obtained with no use of GPT-3 at all. Additionally, we tried using Flan-T5 XL instead of Flan-T5 XXL for the second stage of synthetic query generation, since it is more than 90% smaller than Flan-T5 XXL in terms of model parameters. This still leads to better performance than Zero-shot ColBERTv2.

We also explored using a smaller cross-encoder

for UDAPDR. We tested using DeBERTaV3-Base instead of DeBERTaV3-Large for our passage reranker, decreasing the number of model parameters by over 70%. We found that DeBERTaV3-Base was still effective, though it results in a 4.1 point drop in Success@5 compared to DeBERTaV3-Large for LoTTE Pooled (Table 3). (In our initial explorations, we also tested using BERT-Base or RoBERTa-Large as the cross-encoder but found them less effective than DeBERTaV3, leading to 6–8 point drops in Success@5.)

## 4.6 Different Prompting Strategies

We tested whether a simpler few-shot prompting strategy might be better than our corpus-adapted prompting approach for domain adaptation. In Table 4, we compare the InPars (Bonifacio et al., 2022) few-shot prompt to our corpus-adapted prompt approach for synthetic query generation and passage reranker distillation. We evaluate these using query generation with both Flan XXL and GPT-3. We find that our multi-reranker, corpus-

| Query Generators | Passage Reranker | Success@5 |
|---|---|---|
| GPT-3 + Flan-T5 XXL | DeBERTav3 Large | **71.1** |
| GPT-3 + Flan-T5 XL | DeBERTav3 Large | 66.7 |
| Flan-T5 XXL | DeBERTav3 Large | 68.0 |
| Flan-T5 XL | DeBERTav3 Large | 65.9 |
| GPT-3 + Flan-T5 XXL | DeBERTav3 Base | 67.0 |
| GPT-3 + Flan-T5 XL | DeBERTav3 Base | 64.1 |
| Zero-shot ColBERTv2 | N/A | 63.7 |

Table 3: **Model Configurations for Synthetic Query Generation and Passage Reranker.** We describe the first and second query generators for UDAPDR in Section 3. The Success@5 scores are for the LoTTE Pooled dev task. A single non-distilled reranker is trained on 100K synthetic queries for each configuration. We do not explore a configuration with exclusively GPT-3 generated queries due to GPT-3 API costs.

| Prompt Strategy | Query Generators | Reranker Count | Success@5 |
|---|---|---|---|
| InPars Prompt | GPT-3 | 1 | 65.8 |
| InPars Prompt | Flan-T5 XXL | 1 | 67.6 |
| InPars Prompt | Flan-T5 XXL | 5 | 67.1 |
| Corpus-Adapted Prompts | GPT-3 + Flan-T5 XXL | 1 | 67.4 |
| Corpus-Adapted Prompts | GPT-3 + Flan-T5 XXL | 5 | **71.1** |
| Zero-shot ColBERTv2 | N/A | N/A | 63.7 |

Table 4: **Model Configurations for Prompting Strategies.** We specify the prompting strategy, query generators, and reranker counts for each configuration. The Success@5 scores are for the LoTTE Pooled dev task. 100,000 synthetic queries total are used for each approach except for the top row, which uses 5,000 synthetic queries due to the costs of the GPT-3 API. The synthetic queries are split evenly amongst the total rerankers used. The rerankers are distilled with a ColBERTv2 retriever for configuration.

adapted prompting strategy is more successful, leading to a 3.5 point increase in Success@5 after ColBERTv2 distillation while using the same number of synthetic queries for training.

## 4.7 LoTTE and BEIR Test Results

In Table 5 and Table 6, we include the test set results for LoTTE and BEIR, respectively. For LoTTE, UDAPDR increases ColBERTv2 zero-shot Success@5 for both Forum queries and Search queries, leading to a 7.1 point and a 3.9 point average improvement, respectively. For BEIR, we calculated ColBERTv2 accuracy using nDCG@10. We found that UDAPDR increases zero-shot accuracy by 5.2 points on average. Promptagator++ Few-shot offers similar improvements to zero-shot accuracy, achieving a 4.2 point increase compared to a zero-shot ColBERTv2 baseline. However, Promptagator++ Few-shot also uses a reranker during retrieval, leading to additional computational costs at inference time. By comparison, UDAPDR is a zero-shot method (i.e., that does not assume access to gold labels from the target domain) and only uses the ColBERTv2 retriever and thus has a lower query latency at inference time.

## 4.8 Additional Results

Table 1 and Table 2 explore only a limited range of potential uses for UDAPDR. In Appendix A, we consider a wider range of uses. First, we ask whether it is productive to filter the set of rerankers based on in-domain dev set performance. We mostly find that this does not lead to gains over simply using all of them, and it introduces the requirement that we have a labeled dev set. Second, we evaluate whether substantially increasing the value of $Z$ from 100K leads to improvements. We find that it does not, and indeed that substantially larger values of $Z$ can hurt performance.

## 5 Discussion & Future Work

Our experiments with UDAPDR suggest several directions for future work:

- While we show that our domain adaptation strategy is effective for the multi-vector ColBERTv2 model, whether it is also effective for other retrieval models is an open question for future research.

- For our base model in ColBERTv2, we use BERT-Base. However, ColBERTv2 now allows for other base models, such as DeBER-TaV3, ELECTRA (Clark et al., 2020), and

| | LoTTE Datasets | | | | | | | | | | | |
| | **Forum** | | | | | | **Search** | | | | | |
| | Life. | Tech. | Writing | Rec. | Science | Pooled | Life. | Tech. | Writing | Rec. | Science | Pooled |
|---|---|---|---|---|---|---|---|---|---|---|---|---|
| BM25 | 60.6 | 39.4 | 64.0 | 55.4 | 37.1 | 47.2 | 63.8 | 41.8 | 60.3 | 56.5 | 32.7 | 48.3 |
| SPLADEv2 | 74.0 | 50.8 | 73.0 | 67.1 | 43.7 | 60.1 | 82.3 | 62.4 | 77.1 | 69.0 | 55.4 | 68.9 |
| RocketQAv2 | 73.7 | 47.3 | 71.5 | 65.7 | 38.0 | 57.7 | 82.1 | 63.4 | 78.0 | 72.1 | 55.3 | 69.8 |
| Zero-shot ColBERTv2 | 76.2 | 54.0 | 75.8 | 69.8 | 45.6 | 62.3 | 82.4 | 65.9 | 80.4 | 73.2 | 57.5 | 71.5 |
| UDAPDR | **84.9** | **59.9** | **83.2** | **78.6** | **48.8** | **70.8** | **86.8** | **67.7** | **84.3** | **77.9** | **61.0** | **76.6** |

Table 5: **Success@5 for LoTTE Test Set Results.** The ColBERTv2 distillation strategies train $Y$ rerankers each with $Z$ synthetic queries before distilling each reranker with the same ColBERTv2 model. For UDAPDR, we use 5 rerankers and 20K distinct synthetic queries for training each of them. For BM25, SPLADEv2, and RocketQAv2, we take results directly from Santhanam et al. (2022b).

| | BEIR Datasets | | | | | | | | | | |
| | ArguAna | Touché | Covid | NFcorpus | HotpotQA | DBPedia | Climate-FEVER | FEVER | SciFact | SCIDOCS | FiQA |
|---|---|---|---|---|---|---|---|---|---|---|---|
| BM25 | 31.5 | **36.7** | 65.6 | 32.5 | 60.3 | 31.3 | 21.3 | 75.3 | 66.5 | 15.8 | 23.6 |
| DPR (MS MARCO) | 41.4 | - | 56.1 | 20.8 | 37.1 | 28.1 | 17.6 | 58.9 | 47.8 | 10.8 | 27.5 |
| ANCE | 41.5 | - | 65.4 | 23.7 | 45.6 | 28.1 | 19.8 | 66.9 | 50.7 | 12.2 | 29.5 |
| ColBERT (v1) | 23.3 | - | 67.7 | 30.5 | 59.3 | 39.2 | 18.4 | 77.1 | 67.1 | 14.5 | 31.7 |
| TAS-B | 42.7 | - | 48.1 | 31.9 | 58.4 | 38.4 | 22.8 | 70.0 | 64.3 | 14.9 | 30.0 |
| RocketQAv2 | 45.1 | 24.7 | 67.5 | 29.3 | 53.3 | 35.6 | 18.0 | 67.6 | 56.8 | 13.1 | 30.2 |
| SPLADEv2 | 47.9 | 27.2 | 71.0 | 33.4 | 68.4 | 43.5 | 23.5 | 78.6 | 69.3 | 15.8 | 33.6 |
| BM25 Reranking w/ Cohere$_{large}$ | 46.7 | 27.6 | 80.1 | 34.7 | 58.0 | 37.2 | 25.9 | 67.4 | 72.1 | **19.4** | 41.1 |
| BM25 Reranking w/ OpenAI$_{ada2}$ | 56.7 | 28.0 | 81.3 | **35.8** | 65.4 | 40.2 | 23.7 | 77.3 | **73.6** | 18.6 | 41.1 |
| Zero-shot ColBERTv2 | 46.1 | 26.3 | 84.7 | 33.8 | 70.3 | 44.6 | 27.1 | 78.0 | 66.0 | 15.4 | 45.8 |
| GenQ | 49.3 | 18.2 | 61.9 | 31.9 | 53.4 | 32.8 | 17.5 | 66.9 | 64.4 | 14.3 | 30.8 |
| GPL + TSDAE | 51.2 | 23.5 | 74.9 | 33.9 | 57.2 | 36.1 | 22.2 | 78.6 | 68.9 | 16.8 | 34.4 |
| UDAPDR | **57.5** | 32.4 | **88.0** | 34.1 | **75.3** | **47.4** | **33.7** | **83.2** | 72.2 | 17.8 | **53.5** |
| Promptagator Few-shot | 63 | 38.1 | 76.2 | 37 | 73.6 | 43.4 | 24.1 | 86.6 | 73.1 | 20.1 | 49.4 |

Table 6: **nDCG@10 for BEIR Test Set Results.** For each dataset, the highest-accuracy *zero-shot* result is marked in bold while the highest overall is underlined. For UDAPDR, we use 5 rerankers and 20K distinct synthetic queries for training each of them. The Promptagator and GPL results are taken directly from their respective papers. For Promptagator, we include both the best retriever-only configuration (Promptagator Few-shot) and the best retriever + reranker configuration (Promptagator++ Few-shot). We include the GPL+TSDAE pretraining strategy, which is found to improve retrieval accuracy (Wang et al., 2022). We copy the results for BM25, GenQ, ANCE, TAS-B, and ColBERT from Thakur et al. (2021), for MoDIR and DPR-M from Xin et al. (2022), for SPLADEv2 from Formal et al. (2021), and for BM25 Reranking of Cohere$_{large}$ and OpenAI$_{ada2}$ from Kamalloo et al. (2023).

RoBERTa (Liu et al., 2019). We would be interested to see the efficacy of our domain adaptation strategy with these larger encoders.

- We explored distillation strategies for combining passage rerankers with ColBERTv2. However, testing distillation strategies for shrinking the reranker itself could be a viable direction for future work.

- We draw upon several recent publications, including Bonifacio et al. (2022) and Asai et al. (2022), to create the prompts used for GPT-3 and Flan-T5 XXL in our domain adaptation strategy. Creating a more systematic approach for generating the initial prompts would be an important item for future work.

# 6   Conclusion

We present UDAPDR, a novel strategy for adapting retrieval models to new domains. UDAPDR uses synthetic queries created using generative models, such as GPT-3 and Flan-T5 XXL, to train multiple passage rerankers on queries for target domain passages. These passage rerankers are then distilled into ColBERTv2 to boost retrieval accuracy while keeping query latency competitive as compared to other retrieval systems. We validate our approach across the LoTTE, BEIR, NQ, and SQuAD datasets. Additionally, we explore various model configurations that alter the generative models, prompting strategies, retriever, and passage rerankers used in our approach. We find that UDAPDR can boost zero-shot retrieval accuracy on new domains without the use of labeled training examples. We also discuss several directions for future work.

## 7 Limitations

While our domain adaptation technique does not require questions or labels from the target domain, it does require a significant number of passages in the target domain. These passages are required for use in synthetic query generation with the help of LLMs like GPT-3 and Flan-T5, so future work should evaluate how effective these methods are on extremely small passage collections.

The synthetic queries created in our approach may also inherit biases from the LLMs and their training data. Moreover, the exact training data of the LLMs is not precisely known. Our understanding is that subsets of SQuAD and NQ, in particular, have been used in the pretraining of Flan-T5 models as we note in the main text. More generally, other subtle forms of data contamination are possible as with all research based on LLMs that have been trained on billions of tokens from the Web. We have mitigated this concern by evaluating on a very large range of datasets and relying most heavily on open models (i.e., Flan-T5, DeBERTa, and ColBERT). Notably, our approach achieves consistently large gains across the vast majority of the many evaluation datasets tested (e.g., the individual sets within BEIR), reinforcing our trust in the validity and transferability of our findings.

Additionally, the LLMs used in our technique benefit substantially from GPU-based hardware with abundant and rapid storage. These technologies are not available to all NLP practitioners and researchers due to their costs. Lastly, all of our selected information retrieval tasks are in English, a high-resource language. Future work can expand on the applicability of our domain adaptation techniques by using non-English passages in low-resource settings, helping us better understand the approach's strengths and limitations.

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

## Appendix

## A Reranker Configurations

We aim to understand the impact of different model configurations on the efficacy of our domain adaptation technique. We expand on experiments from Section 4.3 and explore alternate configurations of key factors in the UDAPDR approach. Specifically, we want to answer the following questions through their corresponding experiments:

1. (a) **Question**: Do corpus-adapted prompts improve domain adaptation and retriever distillation?
   (b) **Experiment**: Compare zero/few-shot prompts to corpus-adapted prompts for synthetic question generation (Table 4).

2. (a) **Question**: How does the number of rerankers affect downstream retrieval accuracy?
   (b) **Experiment**: Compare single-reranker to multi-reranker approach (with different reranker selection strategies) across various target domains (Table 7).

3. (a) **Question**: How does the synthetic query count affect downstream retrieval accuracy?
   (b) **Experiment**: Explore different query count configurations ranging from several thousand to hundreds of thousands of synthetic queries (Table 1).

4. (a) **Question**: How do triple counts during distillation affect domain adaptation for ColBERTv2?
   (b) **Experiment**: Explore different triple counts ranging from several thousand to millions of triples (Table 8).

In Table 7 and Table 8 (and Table 1 in the main text), we outline the results of different experimental configurations in which we alter synthetic query generation and passage reranker training. Based on our results for the LoTTE pooled dataset, we find that training multiple rerankers and selecting the best performing rerankers can improve our unsupervised domain adaptation approach. We generate multiple corpus-adapted prompts and rerankers to prevent edge cases in which sampled in-domain passages and queries have low quality. Furthermore, distilling the passage rerankers with ColBERTv2 allows us preserve their accuracy gains while avoiding their high computational costs. However, training many rerankers and selecting the best five rerankers can be computationally intensive and ultimately unnecessary to achieve domain adaptation. The simpler approach of training several rerankers and using them for distillation, without any quality filtering, yields comparable results with only a 0.6 point drop in Success@5 on average while computing 10x less synthetic queries (Table 7). Additionally, by using our rerankers to generate more triples for distillation with ColBERTv2, we were able to boost performance even further as shown in Table 8.

## B Fine-tuning Rerankers and Retriever

For all passage reranker models that we fine-tune, we optimize for cross-entropy loss using Adam (Kingma and Ba, 2014) and apply a 0.1 dropout to the Transformer outputs. We feed the final hidden state of the [CLS] token into a single linear classification layer. We fine-tune for 1 epoch in all experimental configurations. Additionally, we using a 5e-6 learning rate combined with a linear warmup and linear decay for training (Howard and Ruder, 2018). We use a batch size of 32 across all experimental configurations.

For our ColBERTv2 retriever, we use a 1e-5 learning rate and a batch size of 32 during distillation. The ColBERTv2 maximum document length is set to 300 tokens. We use a BERT-Base model (Devlin et al., 2019) as our encoder.

Instead of fine-tuning the rerankers, we also tried fine-tuning ColBERTv2 directly with the synthetic datasets. We found that fine-tuning the retriever directly with the synthetic queries offered only limited benefits, only improving zero-shot retrieval by 1-3 points of accuracy at best and decreasing zero-shot accuracy at worst (for the LoTTE Forum dev set). Distilling the rerankers offered more substantive gains and better adaptation to the target domains more generally.

| | ColBERTv2 Distillation with UDAPDR | | | | |
|---|---|---|---|---|---|
| | Zero-shot ColBERTv2 | 1 of 50 Rerankers | 5 of 50 Rerankers | 10 of 50 Rerankers | 5 of 5 Rerankers | Zero-shot ColBERTv2 + Reranker |
| LoTTE Lifestyle | 64.5 | 67.5 | 73.2 | **73.6** | 72.8 | 73.5 |
| LoTTE Techology | 44.5 | 46.3 | 50.3 | **50.7** | 49.9 | 50.6 |
| LoTTE Writing | 80.0 | 81.5 | 83.4 | 84.2 | 83.0 | **85.5** |
| LoTTE Recreation | 70.8 | 73.7 | 77.8 | 78.3 | 76.7 | **79.1** |
| LoTTE Science | 61.5 | 63.0 | 66.3 | 66.8 | 65.5 | **67.2** |
| LoTTE Pooled | 63.7 | 66.3 | 70.3 | 70.7 | 69.6 | **71.1** |
| NaturalQuestions | 68.9 | 70.1 | 73.0 | 73.5 | 72.7 | **73.9** |
| SQuAD | 65.0 | 67.2 | 71.0 | 71.3 | 70.4 | **72.6** |

Table 7: **Success@5 for Multi-Reranker Domain Adaptation Strategies with Different Reranker Counts.** LoTTE dataset results correspond to the Forum configuration. All results correspond to dev sets of each task. The reranker used in the experiments is DeBERTa-v3-Large. The ColBERTv2 distillation strategies train $X$ number of rerankers before selecting the best $Y$ based on their performance on the dev set of the target domain. Through the selection process, we aim to find an upper bound for retrieval accuracy, even though access to an annotated dev set is not realistic for all domains. In our distillation strategies, each reranker was trained on 2,000 synthetic queries. For our non-distilled reranker used in the final column, we trained it on 100,000 synthetic queries. The synthetic queries were created using Flan-T5 XXL model and the prompting strategy outlined in Section 3.

| | ColBERTv2 Distillation with UDAPDR | | | |
|---|---|---|---|---|
| | Zero-shot ColBERTv2 | 1000 triples | 10000 triples | 100000 triples | Zero-shot ColBERTv2 + Reranker |
| LoTTE Lifestyle | 64.5 | 67.4 | 70.1 | 72.4 | **73.5** |
| LoTTE Techology | 44.5 | 46.0 | 47.8 | 50.2 | **50.6** |
| LoTTE Writing | 80.0 | 82.5 | 83.4 | 85.0 | **85.5** |
| LoTTE Recreation | 70.8 | 72.3 | 76.5 | 77.7 | **79.1** |
| LoTTE Science | 61.5 | 62.0 | 64.2 | 66.5 | **67.2** |
| LoTTE Pooled | 63.7 | 66.0 | 68.4 | 70.4 | **71.1** |
| NaturalQuestions | 68.9 | 70.5 | 72.7 | 73.4 | **73.9** |
| SQuAD | 65.0 | 67.2 | 70.6 | 71.5 | **72.6** |

Table 8: **Success@5 for Multi-Reranker Domain Adaptation Strategies with Various Distillation Triples Counts.** LoTTE dataset results correspond to the Forum configuration. All results correspond to dev sets of each task. The reranker used in the experiments is DeBERTa-v3-Large. The ColBERTv2 distillation strategies use a single reranker trained on 10,000 synthetic queries; this reranker then generates the specified number of labeled triples. For our non-distilled reranker used in the final column, we trained it on 100,000 synthetic queries. The synthetic queries were created using Flan-T5 XXL model and the prompting strategy outlined in Section 3.

We ran additional experiments testing UDAPDR's efficacy on LoTTE Search dev, using one reranker trained with a unique set of 2000 synthetic queries. We found that the approach boosted accuracy by 1.6 points, increasing accuracy from 71.5 to 73.1. However, since the synthetic queries could be generated so cheaply, we decided to scale to tens of thousands of synthetic queries for further experiments.