# OpenReview forum: "UDAPDR: Unsupervised Domain Adaptation via LLM Prompting and Distillation of Rerankers"
_EMNLP/2023/Conference — EMNLP 2023 Main_

### Official Review · Reviewer_LCfk · 2023-08-04

**Soundness:** 4

**Excitement:**

4: Strong: This paper deepens the understanding of some phenomenon or lowers the barriers to an existing research direction.

**Paper Topic And Main Contributions:**

the submission proposed a practical pipeline that 1) leverages a very large language model for generating a small batch of samples, and then 2) uses a small language model to enrich the training samples, and  3) train multiple rerankers and distill them into a retrieval model. Although the pipeline seems rather tedious, the performance has improved significantly in comparison to baseline approaches. Ablation study also shows the necessity of individual components.

**Reasons To Accept:**

1. the pipeline takes the inference latency of very large language models into consideration, and only uses it for the very first stage.

2. the pipeline then  reduces the inference latency of multiple rerankers by distilling them into a single retrieval model.

3. ablation study is thorough, and addresses common questions regarding the use of large language models.

**Reasons To Reject:**

1. it seems very interesting that by using GPT-3 alone, the performance is already better than zero-shot approaches. it raises the question on the future of this approach since hardware devices will eventually catch up, and using GPT-3 won't be a very time-consuming task anymore.

2. i am slightly curious about the total computational cost of training the entire pipeline. There are many recent work on parameter-efficient finetuning and also distillation that potentially also allow you to directly obtain a small model for fast inference at the end. Any of recent finetuning and/or distillation work on top of GPT-3 or FlanT5XXL would be a very strong competitor. Could the authors comment on this?

**Reproducibility:**

4: Could mostly reproduce the results, but there may be some variation because of sample variance or minor variations in their interpretation of the protocol or method.

**Reviewer Confidence:**

4: Quite sure. I tried to check the important points carefully. It's unlikely, though conceivable, that I missed something that should affect my ratings.

---

> ### Author Rebuttal · Authors · 2023-08-28
>
> Thank you for the feedback on our submission! We are glad that you appreciate our ablation study and the practicality of the UDAPDR fine-tuning pipeline, particularly when it comes to reducing the high latency costs of LLMs. We wanted to address your concerns about the use of GPT-3 and the total computational pipeline for UDAPDR:
>
> > “it seems very interesting that by using GPT-3 alone, the performance is already better than zero-shot approaches. it raises the question on the future of this approach since hardware devices will eventually catch up, and using GPT-3 won't be a very time-consuming task anymore.”
>
> UDAPDR offers a general two-stage method for domain adaptation. If GPT-3 becomes a little cheaper, UDAPDR can be configured to make wider use of it, especially in smaller domains. More fundamentally, though, we postulate that the tradeoff between scale (and hence cost) and quality will continue to exist with newer and larger models. (Think GPT-4, GPT-5, etc. so to speak.)
>
> Because of this, our two-stage approach will continue to have value for scaling down the quality of the strongest LLMs for synthetic query generation so it’s feasible over large corpora.
> In addition, even if large language models like GPT-3 and FLAN-T5 XXL could be hosted locally and used for reranking, it would be unlikely that their latency costs would be conducive for user-facing settings of information retrieval. Therefore, distillation of rerankers would still be required at some stage to preserve competitive latency at inference.
>
>
> > “i am slightly curious about the total computational cost of training the entire pipeline. There are many recent work on parameter-efficient finetuning and also distillation that potentially also allow you to directly obtain a small model for fast inference at the end. Any of recent finetuning and/or distillation work on top of GPT-3 or FlanT5XXL would be a very strong competitor. Could the authors comment on this?”
>
> Overall, the computational pipeline for UDAPDR ranges from 3 to 6 hours, with additional training time further improving the accuracy and distillation gains of our approach.  For comparison: fine-tuning GPT-3.5 using existing frameworks takes several days when including data validation, uploading, training, and deployment. Parameter-efficient fine-tuning (PEFT) and reduced precision configurations for FLAN-T5 XXL could lead to reduced quality for synthetic query generation in UDAPDR, substantially decreasing the efficacy of our technique, but we could consider them for scenarios where 3-6 hours is too long.
>
>
> Finally, we want to mention that UDAPDR will be released as part of an open-source package for doing retrieval and OpenQA. This release will allow the reproduction of the results in the paper, and its primary goal is to make UDAPDR useful for users in general.

---

### Official Review · Reviewer_8wrV · 2023-08-05

**Typos Grammar Style And Presentation Improvements:** 1. line 019 2k -> 20k?
**Soundness:** 3

**Excitement:**

3: Ambivalent: It has merits (e.g., it reports state-of-the-art results, the idea is nice), but there are key weaknesses (e.g., it describes incremental work), and it can significantly benefit from another round of revision. However, I won't object to accepting it if my co-reviewers champion it.

**Paper Topic And Main Contributions:**

The paper proposed an unsupervised domain adaptation approach using LLM for synthetic data generation and multiple reranker distillation, called UDAPDR. The proposed method includes 3 stages for synthetic data generation, including generating synthetic queries on a few target domain passages, converting the generated queries into corpus-adapted prompts, and generating large scale synthetic data using the prompts. Then multiple rerankers are trained using the generated set and are then distilled to a single ColBERTv2 retriever. The distilled ColBERTv2 model is efficient and maintain the performance of rerankers. The evaluation on several datasets show the effectiveness of UDAPDR.

**Questions For The Authors:**

1) In section 4.5, when not using GPT3 at all, does it mean Flan-T5 is used in stage 1?

**Reasons To Accept:**

The paper proposed a domain adaptation approach that doesn't require any labeled data and a small amount of synthetic data. The proposed method is easy but effective. Extensive evaluation and ablation experiments show the effectiveness of UDAPDR.

**Reasons To Reject:**

The major concern is that the proposed approach is not novel. The prompts used in stage 1 and 2 are seen from prior works.

Also, compared with few-shot method proposed in promptagator, this paper uses GPT3 to generate corpus-adapted prompt instead of manually select a few examples in promptagator. It is not clear if the proposed approach is more effective and less cost than promptagator. For example, not all tasks in BEIR are question answering, but the prompts used in stage 1 are focused on generating question given document. It seems that it would be difficult to adapt to a different task, e.g. counter argument retrieval in ArguAna.

In addition, it would be interesting to know the performance of using the generated synthetic set to train a first-stage retrieval model as well.

**Reproducibility:**

3: Could reproduce the results with some difficulty. The settings of parameters are underspecified or subjectively determined; the training/evaluation data are not widely available.

**Reviewer Confidence:**

4: Quite sure. I tried to check the important points carefully. It's unlikely, though conceivable, that I missed something that should affect my ratings.

---

> ### Author Rebuttal · Authors · 2023-08-28
>
> Thank you for the feedback on our submission! We are happy to see you recognize our in-depth evaluations and ablation studies for UDAPDR and are excited to see that you found it “easy but effective” as is our goal for this work. We hope the experiments illustrate the utility of synthetic query generation and the efficacy of our reranker distillation technique. Here, we wanted to address your concerns about the paper’s contributions:
>
> > “The major concern is that the proposed approach is not novel. The prompts used in stage 1 and 2 are seen from prior works.”
>
> Only the initial, manual prompts used in Stage #1 are from prior work while the prompts used in Stage #2 are unique and corpus-adapted for each domain. The heart of our approach is a new technique for creating integrated retrievers that are robust across domains. For this, we leverage proven prompting strategies to avoid engaging in unnecessary prompt hacking. This is simply an experimental choice for UDAPDR, not the essence of the method. Overall, we use the queries to train rerankers before distilling them with the ColBERTv2 retriever, bolstering domain adaptation while preserving a competitive retrieval latency at inference.
>
>
> > “Also, compared with few-shot method proposed in promptagator, this paper uses GPT3 to generate corpus-adapted prompt instead of manually select a few examples in promptagator. It is not clear if the proposed approach is more effective and less cost than promptagator.”
>
> The evidence in the paper strongly supports the claim that UDAPDR is more efficient and effective than Promptagator, GPL, and the other domain adaptation techniques we evaluate. The most successful configuration of Promptagator is Promptagator++, which combines a fine-tuned retriever and reranker. The approach requires several human-annotated examples and utilization of LLMs like FLAN-T5 for synthetic data generation, achieving a 6.2 point increase in nDCG@10 on average for BEIR (46.6 → 52.8). However, their reranker introduces a high cost at inference, which they do not quantify in the paper but recognize as a relevant challenge for future research. Without their reranker, they only achieve 1.2 point increase in nDCG@10 on average for BEIR.
>
> In comparison, UDAPDR does not use any rerankers at inference, instead distilling them with the ColBERTv2 retriever to preserve competitive latency for users while achieving a 5.2 point increase in nDCG@10 on average for BEIR. Additionally, UDAPDR does not require any human-annotated examples; it also utilizes FLAN-T5 and offers the option to use GPT-3 for higher-quality synthetic queries. Overall, the accuracy gains and greater inference efficiency of UDAPDR give our approach an edge over Promptagator
>
>
> > “ For example, not all tasks in BEIR are question answering, but the prompts used in stage 1 are focused on generating question given document. It seems that it would be difficult to adapt to a different task, e.g. counter argument retrieval in ArguAna.”
>
> UDAPDR still proves effective for BEIR tasks not centered around question-answering by helping adapt the ColBERTv2 retriever towards the in-domain documents. Table 6 reports a full evaluation, which includes an outstanding score on ArguAna – the best zero-shot score. Our two-stage synthetic query generation approach combined with query filtering allows us to generate relevant query/document pairs for fine-tuning rerankers in the new domain. As a result, after distilling our rerankers with the retriever, our retrievers are much better equipped to handle the specialized domains in BEIR. Nonetheless, we fully agree that it is possible to obtain an even more general version of UDAPDR by improving the seed set of prompts. We will urge the community to consider such direct extensions for their own applications.
>
>
> > “In addition, it would be interesting to know the performance of using the generated synthetic set to train a first-stage retrieval model as well.”
>
> We found that fine-tuning the retriever directly with the synthetic queries offered only limited benefits, only improving zero-shot retrieval by 1-3 points of accuracy at best and decreasing zero-shot accuracy at worst (for the LoTTE Forum dev set). Distilling the rerankers offered more substantive gains and better adaptation to the target domains more generally. We will include the retriever fine-tuning experiments in our revised paper.
>
> Additionally, we wanted to address each of your questions:
>
> Yes, in section 4.5 when we don’t use GPT-3 at all, we use FLAN-T5 in stage #1 instead.
>
> Also, thank you for pointing out the grammatical errors and presentational improvements! We will address each of them in our revised paper.
>
> Finally, we want to mention that UDAPDR will be released as part of a popular open-source package for doing retrieval and OpenQA. This release will allow the reproduction of the results in the paper, and its primary goal is to make UDAPDR useful for users in general.

---

### Official Review · Reviewer_1Fo5 · 2023-08-12

**Typos Grammar Style And Presentation Improvements:** 1. I found the dotted arrow in Figure…
**Soundness:** 3

**Excitement:**

4: Strong: This paper deepens the understanding of some phenomenon or lowers the barriers to an existing research direction.

**Paper Topic And Main Contributions:**

This paper presents UDAPDR, a novel domain adaptation method for retrieval models. This method leverages an effective LLM to generate queries of the target domain, which are then used to prompt a less expensive LLM to generate more queries to train a reranker. Last, this reranker is distilled into a final retriever. The authors tested their method on LoTTE and BEIR.

**Questions For The Authors:**

1. How does the choice of prompts in Stage 1 affect the overall performance? What if only a subset of them are used?
2. The prompt template in Stage 2 includes both good and bad questions. However, only Prompt 1 in Stage 1 generates bad questions. Could you clarify how bad questions are generated for other prompts?
3. What if the generated queries from stage 3 are used for training the retriever directly instead of training a reranker and then distilling?
4. What's the total cost of running GPT-3 at the first stage?

**Reasons To Accept:**

The proposed method manages to improve the out-of-distribution generation of neural retrievers without adding latencies at the test time. The adaptation improvement is consistent across multiple datasets (BEIR & BEIR). This method is more computationally efficient compared to previous adaptation methods using synthetic queries.

**Reasons To Reject:**

1. I don't see the claim in line 107-108, "our technique only requires 1000s of synthetic queries to prove effective" is well supported by the experiment.
2. Although Flan-T5 XXL is much smaller than GPT-3, it is a large model and its computational cost is not negligible.
3. Please see the Questions For The Authors section.

**Reproducibility:**

4: Could mostly reproduce the results, but there may be some variation because of sample variance or minor variations in their interpretation of the protocol or method.

**Reviewer Confidence:**

3: Pretty sure, but there's a chance I missed something. Although I have a good feel for this area in general, I did not carefully check the paper's details, e.g., the math, experimental design, or novelty.

---

> ### Author Rebuttal · Authors · 2023-08-28
>
> Thank you for the feedback on our submission! We are glad that appreciate about our work that it “manages to improve the out-of-distribution generation of neural retrievers without adding latencies”, that its “improvement is consistent across multiple datasets”, and that it’s “more computationally efficient compared to previous adaptation methods using synthetic queries”. We wanted to address your concerns about the paper’s contributions:
>
> > “I don't see the claim in line 107-108, "our technique only requires 1000s of synthetic queries to prove effective" is well supported by the experiment.”
>
> Thank you for noting this. The paper did previously operate mainly at the level of 10K examples. To justify this claim, we ran a new experiment testing UDAPDR’s efficacy on LoTTE Search dev, using one reranker trained with a unique set of 2000 synthetic queries. We found that the approach boosted accuracy by 1.6 points, increasing accuracy from 71.5 to 73.1. We will include this result and expand it to more settings for the final paper, as it provides important new insights into the method.
>
>
> > “Although Flan-T5 XXL is much smaller than GPT-3, it is a large model and its computational cost is not negligible.”
>
> FLAN-T5 XXL requires significant resources for running inference at full-precision, though that depends on the prompt length. In Table 3, we explore the success of using FLAN-T5 XL (90% smaller than XXL) instead for synthetic generation. Furthermore, we plan to evaluate reduced precision configurations for FLAN-T5 XXL in our revised paper.
>
> Additionally, we wanted to address each of your questions:
>
> > “How does the choice of prompts in Stage 1 affect the overall performance? What if only a subset of them are used?”
>
> We did not perform extensive prompt engineering experiments for stage #1 since we found the prompts selected to be effective enough for our approach. From initial exploration, we found that similar prompt configurations were not more effective than the tested prompts. Testing extensive prompt configurations might also lead to an excessive number of experimental configurations that was too large for our available resources.
>
>
> > “The prompt template in Stage 2 includes both good and bad questions. However, only Prompt 1 in Stage 1 generates bad questions. Could you clarify how bad questions are generated for other prompts?”
>
> To generate all of our bad questions for the other prompts, we solely used prompt #1. We will make sure to clarify this distinction in the revised paper.
>
>
> > “What if the generated queries from stage 3 are used for training the retriever directly instead of training a reranker and then distilling?”
>
> We found that fine-tuning the retriever directly with the synthetic queries offered only limited benefits, only improving zero-shot retrieval by 1-3 points of accuracy at best and decreasing zero-shot accuracy at worst (for the LoTTE dev set). Distilling the rerankers offered more substantive gains and better adaptation to the target domains more generally. We will include the retriever fine-tuning experiments in our revised paper.
>
>
> > “What's the total cost of running GPT-3 at the first stage?”
>
> Depending on the context lengths of the target domain, the cost of running GPT-3 for the first stage was ~$0.25-1 (for each run per domain) from November 2022 to March 2023. It is approximately the same cost in August 2023.
>
> Also, thank you for pointing out the grammatical errors and presentational improvements! We will address each of them in our revised paper.
>
> Finally, we want to mention that UDAPDR will be released as part of an open-source package for doing retrieval and OpenQA. This release will allow the reproduction of the results in the paper, and its primary goal is to make UDAPDR useful for users in general.

---

### Official Review · Reviewer_H5VJ · 2023-08-13

**Soundness:** 3

**Excitement:**

3: Ambivalent: It has merits (e.g., it reports state-of-the-art results, the idea is nice), but there are key weaknesses (e.g., it describes incremental work), and it can significantly benefit from another round of revision. However, I won't object to accepting it if my co-reviewers champion it.

**Paper Topic And Main Contributions:**

This paper addresses the problem of domain adaptation in neural information retrieval (IR) models. The authors propose a method called Unsupervised Domain Adaptation via LLM Prompting and Distillation of Rerankers (UDAPDR) to facilitate efficient adaptation of neural retriever models to new domains.

The main contributions of this paper are as follows:

1. UDAPDR: The authors introduce UDAPDR as an unsupervised domain adaptation method for neural IR. It leverages both expensive language models (LLMs) like GPT-3 and less expensive ones like Flan-T5 XXL, along with multiple passage rerankers. This approach aims to improve retrieval accuracy in zero-shot settings for different domains while maintaining competitive latency.

2. Efficient Query Generation: Unlike previous domain adaptation approaches that require millions of synthetic queries, UDAPDR proves effective with only thousands of synthetic queries. The method is compatible with various LLMs designed for instruction-based tasks like query generation. By using multiple prompting strategies, the authors enhance the effectiveness of unsupervised domain adaptation by allowing fine-tuning of multiple rerankers and more effective distillation.

**Reasons To Accept:**

(1) Overall, the idea of synthesizing augmented queries with two-staged LLMs is promising.  Efficiency issues are raised by the authors and relieved by a simple strategy like locate-and-refine that unleashes the power of LLMs.

(2) The results on the four datasets including IR and QA show the effectiveness of the proposed method.

**Reasons To Reject:**

(1) Based on my understanding, Unsupervised Domain Adaptation (UDA) typically involves transferring knowledge gained from source domains, which have abundant annotated training data, to target domains that have only unlabeled data. However, this paper focuses solely on the target domains and does not utilize any data from the source domains. Consequently, it may not be appropriate to classify it as UDA.

(2) The approach taken by LLMs in addressing the issue of domain divergence remains somewhat perplexing, as these models are typically trained to generate general knowledge without specific fine-tuning for particular domains. The authors have not extensively elucidated this aspect or presented empirical evidence to substantiate their claims.

(3) The experimental results indicate that the majority of performance improvements stem from the utilization of the reranker, a widely adopted component in information retrieval (IR) systems. This somewhat diminishes the significance of leveraging LLMs for addressing domain adaptation challenges.

(4) The paper exclusively provides qualitative results, and it would be more compelling to include additional quantitative examples or visualizations to enhance the persuasiveness of the claims made.

**Reproducibility:**

4: Could mostly reproduce the results, but there may be some variation because of sample variance or minor variations in their interpretation of the protocol or method.

**Reviewer Confidence:**

4: Quite sure. I tried to check the important points carefully. It's unlikely, though conceivable, that I missed something that should affect my ratings.

---

> ### Author Rebuttal · Authors · 2023-08-28
>
> Thank you for the feedback on our submission! We are delighted that you found that our key idea can help “unleash the power of LLMs” and that our results “show [its] effectiveness”.
> We’d like to clarify that only the domain-adapted copy of our neural retriever, ColBERTv2, is used at inference; there is no role for rerankers in the final UDAPDR system. Furthermore, our submission provides several quantitative tables as well as a number of figures to illustrate the strengths of UDAPDR compared to other domain adaptation approaches. Here we address your concerns:
>
> > “this paper focuses solely on the target domains and does not utilize any data from the source domains. Consequently, it may not be appropriate to classify it as UDA..”
>
> We share your definition of UDA and would like to clarify where we draw upon source domains for our adaptation technique. For UDAPDR, we leverage knowledge gained from the source domains for the initial retriever, the distilled reranker, and the language models used for synthetic data generation. Our retriever, ColBERTv2, uses MS MARCO training data in addition to the pretraining datasets used for BERT. Our reranker, DeBERTa-v3-Large, additionally has its own host of pretraining datasets, such as the Wikipedia passages. Furthermore, GPT-3 and FLAN-T5 also include massive data collections in their pretraining ensemble. For our domain adaptation approach, we combine the latent knowledge gained from pretraining on the various general domain datasets with in-domain passages to generate synthetic questions and fine-tune our rerankers before distillation with the retriever. Hence, we believe it is fitting to consider UDAPDR a method for domain adaptation (or UDA).
>
>
> > “The approach taken by LLMs in addressing the issue of domain divergence remains somewhat perplexing”
>
> To address domain divergence in our use of LLMs, we prompt the FLAN-T5 and GPT-3 with in-domain passages for generating synthetic questions without the need for in-domain questions or labels. While the synthetic questions do not have the same quality as human-written in-domain questions, they provide sufficient material for adapting our retriever to the specialized domain through the use of reranker distillation. This is a central finding of our work that opens the door to domain transfer, even where human labels are too costly or difficult to obtain.
>
>
> > “The experimental results indicate that the majority of performance improvements stem from the utilization of the reranker”
>
> We must clarify that our final system does not have a reranker but rather is a single efficient ColBERTv2 retrieval model that is used in the target domain (see Figure 1, Stages 5 and 6). Before inference, our trained rerankers are distilled with the retriever, avoiding their costly latency for users while preserving gains to accuracy. As a result, only our neural retriever is used at inference, which has proven competitive in terms of both accuracy and latency. ([Santhanam et al. 2022](https://aclanthology.org/2023.findings-acl.738/) offers a thorough set of recent benchmark numbers.)
>
>
> > “The paper exclusively provides qualitative results, and it would be more compelling to include additional quantitative examples or visualizations to enhance the persuasiveness of the claims made.”
>
> Table 1 through Table 6 are our _quantitative_ results and they amount to a very strong case for UDAPDR compared to other domain adaptive techniques, such as Promptagator and GPL, as well as effective lightweight retrieval techniques, such as BM25 and SPLADEv2. In Table 6, for example, we include a head-on comparison between current state-of-the-art approaches for the BEIR information retrieval benchmark and show where UDAPDR shows significant gains to other neural retrieval approaches. In terms of _qualitative_ results: we feel that the experimental findings yield a very rich picture of system performance and the impact of different design choices across a wide range of task types from community forums to argument retrieval.
> In our revised paper, we would be happy to add a graphical comparison to better visualize our contributions. We will also provide qualitative examples of domains where UDAPDR is most effective (e.g. QA datasets like HotpotQA and FiQA) and domains where UDAPDR barely improves on zero-shot ColBERTv2 (medical/scientific datasets like NFCorpus and SCIDOCS).
>
> Finally, we want to mention that UDAPDR will be released as part of an open-source package for doing retrieval and OpenQA. This release will allow the reproduction of the results in the paper, and its primary goal is to make UDAPDR useful for users in general.

---

### Meta-Review · Area_Chair_PqpY · 2023-09-19

**Recommendation:** 4

**Metareview:**

The authors introduce UDAPDR, a method for adapting neural retriever models to new domains efficiently. UDAPDR leverages both expensive language models (LLMs) like GPT-3 and less expensive ones like Flan-T5 XXL, along with multiple passage rerankers, to improve retrieval accuracy in zero-shot settings while maintaining competitive latency. The contributions of the paper include a novel domain adaptation approach, efficient query generation, and empirical evaluation on datasets like LoTTE and BEIR.


Pros:
The strengths of this paper lie in its innovative approach to domain adaptation, particularly the focus on the target domain without using source domain data.   Reviewers acknowledge that UDAPDR improves out-of-distribution generation of neural retrievers without adding latencies at test time.
 This adaptation improvement is consistent across multiple datasets, demonstrating the method's reliability. The approach is also commended for its computational efficiency compared to previous adaptation methods that use synthetic queries. The paper's contributions, including the pipeline's consideration of inference latency and the comprehensive evaluation and ablation experiments, are highlighted.


Cons:
The computational cost of Flan-T5 XXL is also considered non-negligible. Additionally, questions are raised regarding the novelty of the proposed approach, as certain aspects, such as prompts, have been seen in prior works. The paper is compared to a prior method called "promptagator," and it's suggested that the paper should better demonstrate the effectiveness and cost-efficiency of UDAPDR compared to this method. There is also a suggestion to explore the performance of training a first-stage retrieval model using the generated synthetic set. Lastly, the long-term viability of relying on large language models like GPT-3 for the method is questioned, given the potential for hardware advancements and improved efficiency through other techniques like parameter-efficient finetuning and distillation.
A concern raised by the reviewers  is the lack of strong support for the claim that "only 1000s of synthetic queries".

In summary, the paper presents a promising approach to unsupervised domain adaptation in neural information retrieval models, but concerns have been raised regarding the novelty of the approach, the support for certain claims, and its long-term sustainability with current hardware trends. Addressing these concerns and providing a clearer comparison with existing methods could strengthen the paper's contribution to the field.

---

### Decision · Program_Chairs · 2023-10-07

**Decision:**

Accept-Main

**Comment:**

The authors introduce UDAPDR, a method for adapting neural retriever models to new domains efficiently. UDAPDR leverages both expensive language models (LLMs) like GPT-3 and less expensive ones like Flan-T5 XXL, along with multiple passage rerankers, to improve retrieval accuracy in zero-shot settings while maintaining competitive latency. The contributions of the paper include a novel domain adaptation approach, efficient query generation, and empirical evaluation on datasets like LoTTE and BEIR.


Pros:
The strengths of this paper lie in its innovative approach to domain adaptation, particularly the focus on the target domain without using source domain data.   Reviewers acknowledge that UDAPDR improves out-of-distribution generation of neural retrievers without adding latencies at test time.
 This adaptation improvement is consistent across multiple datasets, demonstrating the method's reliability. The approach is also commended for its computational efficiency compared to previous adaptation methods that use synthetic queries. The paper's contributions, including the pipeline's consideration of inference latency and the comprehensive evaluation and ablation experiments, are highlighted.


Cons:
The computational cost of Flan-T5 XXL is also considered non-negligible. Additionally, questions are raised regarding the novelty of the proposed approach, as certain aspects, such as prompts, have been seen in prior works. The paper is compared to a prior method called "promptagator," and it's suggested that the paper should better demonstrate the effectiveness and cost-efficiency of UDAPDR compared to this method. There is also a suggestion to explore the performance of training a first-stage retrieval model using the generated synthetic set. Lastly, the long-term viability of relying on large language models like GPT-3 for the method is questioned, given the potential for hardware advancements and improved efficiency through other techniques like parameter-efficient finetuning and distillation.
A concern raised by the reviewers  is the lack of strong support for the claim that "only 1000s of synthetic queries".

In summary, the paper presents a promising approach to unsupervised domain adaptation in neural information retrieval models, but concerns have been raised regarding the novelty of the approach, the support for certain claims, and its long-term sustainability with current hardware trends. Addressing these concerns and providing a clearer comparison with existing methods could strengthen the paper's contribution to the field.